# Life-Saver or Undertaker: The Relationship between Primary Cilia and Cell Death in Vertebrate Embryonic Development

**DOI:** 10.3390/jdb10040052

**Published:** 2022-12-12

**Authors:** Thorsten Pfirrmann, Christoph Gerhardt

**Affiliations:** 1Department of Medicine, Health and Medical University, 14471 Potsdam, Germany; 2Institute for Physiological Chemistry, Martin Luther University Halle-Wittenberg, Hollystr. 1, 06144 Halle (Saale), Germany

**Keywords:** apoptosis, transition zone, basal body, axoneme, intraflagellar transport

## Abstract

The development of multicellular organisms requires a tightly coordinated network of cellular processes and intercellular signalling. For more than 20 years, it has been known that primary cilia are deeply involved in the mediation of intercellular signalling and that ciliary dysfunction results in severe developmental defects. Cilia-mediated signalling regulates cellular processes such as proliferation, differentiation, migration, etc. Another cellular process ensuring proper embryonic development is cell death. While the effect of cilia-mediated signalling on many cellular processes has been extensively studied, the relationship between primary cilia and cell death remains largely unknown. This article provides a short review on the current knowledge about this relationship.

## 1. Introduction

Embryonic development depends on a network of closely regulated signalling pathways such as Hedgehog (HH) signalling, Wingless-Int (WNT) signalling, Notch signalling, etc. In vertebrates, these pathways are mediated by tiny, hairlike cellular protrusions referred to as primary cilia. These cellular organelles basically comprise three different structural parts—the basal body (BB), the transition zone (TZ), and the axoneme (Figure 1). The axoneme represents the microtubule-based scaffold of primary cilia. It consists of nine doublet microtubules arranged in a circle and grows out of the BB, a modified centrosome at the ciliary base. The transport of proteins along the axoneme is named intraflagellar transport (IFT). The transport of proteins from the ciliary base to the ciliary tip (anterograde IFT) relies on the motor protein kinesin-2, while the protein transport from the ciliary tip to the ciliary base (retrograde IFT) requires the motor protein dynein-2. During IFT, the cargo proteins are bound to so-called IFT proteins which, in turn, attach to the motor proteins. While the IFT-B complex drives the anterograde IFT, the IFT-A complex is crucial for the retrograde IFT [1]. The proximal end of the axoneme harbours the TZ, a small region controlling ciliary protein import and export. According to its function, it is referred to as the ciliary gatekeeper controlling ciliary protein composition [2]. Ciliary gating is ensured by multiple protein complexes that localise to the TZ [3,4,5,6,7,8,9]. The protein complexes at the TZ comprise the NPHP1-4-8 module, the NPHP5-6 module, the MKS/B9 module and the Inversin (INVS alias NPHP2) compartment [10,11]. These TZ modules are subject to a stringent assembly hierarchy which differs from cell type to cell type [12,13]. In the invertebrate *Caenorhabditis elegans*, Retinitis Pigmentosa GTPase Regulator-Interacting Protein 1-Like (RPGRIP1L alias NPHP8 or MKS5) is in the vanguard of the TZ assembly hierarchy. In vertebrates, the importance of RPGRIP1L in TZ assembly depends on the cell type. In some cell types, RPGRIP1L ensures the proper amount of each TZ module component at the TZ; in other cell types, RPGRIP1L and its vertebrate-specific relative Retinitis Pigmentosa GTPase Regulator-Interacting Protein 1 (RPGRIP1) synergistically regulate the composition of the TZ [12]. Not only TZ assembly but also the appearance of the TZ are cell type specific [14]. For example, in vertebrate photoreceptor cells, the TZ represents the so-called connecting cilium which extends between the outer and inner segment of photoreceptors [15].

Cilia function as signalling hubs mediating numerous signal transduction cascades known to be essential for vertebrate embryonic development. Consequently, the dysfunction of primary cilia causes severe human diseases collectively designated as ciliopathies. Previously, ciliopathies were regarded as rare diseases but, since the number of diseases identified as ciliopathies is permanently rising, this valuation has changed [2]. Recently, it was estimated that ciliopathies affect between 1:700 and 1:2000 people in the general population worldwide [16]. Apart from well-known ciliopathies such as polycystic kidney disease, orofaciodigital syndrome, Meckel–Gruber syndrome, Joubert syndrome, Bardet–Biedl syndrome, Leber congenital amaurosis, Senior–Løken syndrome, Alström syndrome, Jeune asphyxiating thoracic dystrophy, Ellis–van Creveld syndrome, and Sensenbrenner syndrome, the development of more frequent diseases such as cancer or neurodegeneration is related to primary cilia [17,18,19,20,21,22]. In this context, it was shown that primary cilia can either mediate or suppress tumourigenesis depending on the kind of oncogenic initiating event [23]. For example, a mutation in the Hippel–Lindau (VHL) tumour-suppressor gene causes both the loss of renal primary cilia and the development of renal cell carcinomas in humans [24,25]. Importantly, the re-expression of VHL rescues cilia formation [26]. Furthermore, a growing number of studies have revealed a strong link between primary cilia and neurodegenerative diseases, for instance, the essential role of primary cilia in the degeneration of cognitive impairment observed in patients with Alzheimer’s disease by affecting the maturation of cholinergic neurons in the forebrain [22,27].

Importantly, embryogenesis relies on various cellular processes, many of which are regulated by cilia-mediated signalling such as HH signalling, Notch signalling, Transforming Growth Factor-β (TGF-β) signalling, Platelet-Derived Growth Factor Receptor-α (PDGFRα) signalling and Hippo signalling [2,28,29]. These cellular processes comprise cell proliferation, cell differentiation, cell migration, etc. In addition to the listed cellular processes, cell death plays an essential role in embryonic development [30]. One distinguishes between two different types of cell death: accidental cell death (ACD) and regulated cell death (RCD). ACD describes the death of cells as a consequence of severe injury, toxic chemicals, harmful radiation, etc., while RCD is a controlled form of cell death including programmed cell death (PCD) [31]. PCD is not driven by an exogenous environmental disturbance but represents one part of many physiological programmes for development or cell turnover [30,32]. Apoptosis is a type of PCD, characterised by a special cellular ultrastructure comprising condensed chromatin, cytoplasmic compaction and, in some cases, plasma membrane blebbing [33]. Abnormalities in apoptosis cause developmental defects as well as diseases such as cancer and neurodegeneration [30,34,35]. Predominantly, apoptosis is initiated by two different pathways—the extrinsic or death receptor pathway and the intrinsic or mitochondrial pathway (Figure 2).

The extrinsic pathway gets activated by the binding of death ligands [e.g., Tumour Necrosis Factor (TNF), TNF-Related Apoptosis-Inducing Ligand (TRAIL), and FS-7-Associated Surface Antigen (FAS) Ligand] to death receptors [e.g., TNF Receptor 1 (TNFR1), TRAIL Receptor or FAS Receptor]. Upon the formation of the ligand—receptor complex, FAS-Associated Death Domain Protein (FADD) is switched on by Tumour Necrosis Factor Receptor Type 1-Associated Death Domain Protein (TRADD). In turn, FADD activates Pro-Caspase-8 (Pro-CASP8) which then is converted to Caspase-8 (CASP8). Finally, CASP8 induces the effector caspases (CASP3, CASP6, and CASP7) that drive apoptosis.

The intrinsic pathway is stimulated when the cell is exposed to stress stimuli (e.g., lack of growth factors, metabolic stress, loss of adhesion, etc.). It starts with the inhibition of anti-apoptotic members of the B-Cell Lymphoma 2 (BCL-2) protein family [such as BCL-2, Myeloid Cell Leukemia 1 (MCL-1), etc.] by pro-apoptotic members of the same protein family [for example, the BH3 (BCL-2 Homology Domain 3)-only proteins BCL-2-Interacting Mediator of Cell Death (BIM), P53 Upregulated Modulator of Apoptosis (PUMA), BH3-Interacting Domain Death Agonist (BID), BCL-2-Antagonist of Cell Death (BAD), etc.]. This inhibition results in the activation of BCL-2-Associated X Protein (BAX) and BCL-2 Homologous Antagonist/Killer (BAK). Activated BAX and BAK permeabilise the mitochondrial outer membrane leading to the release of Cytochrome-c and further apoptogenic factors from the mitochondria into the cytoplasm. In the cytoplasm, Cytochrome-c, Apoptotic Peptidase Activating Factor 1 (APAF-1) and Caspase-9 (CASP9) form the apoptosome. This formation engenders the activation of CASP9 which subsequently cleaves and, hence, activates the effector caspases CASP3, CASP6, and CASP7 [35,36,37,38,39]. A second CASP9 activation pathway functions independent of the apoptosome and is triggered by increasing ROS concentrations [40].

Although both primary cilia and cell death are of crucial importance for vertebrate embryonic development, their relationship is only poorly understood. In this review, we will summarise and discuss the current knowledge about the relationship between primary cilia and cell death.

## 2. IFT Proteins and Cell Death

In several studies, cell death was analysed in cells and organisms with defective cilia. Some of these investigations provided data suggesting that ciliary dysfunction does not affect cell death. For example, Vion et al. showed that apoptosis is not altered in *Ift88*-negative endothelial cells of developing murine retinas lacking primary cilia. Similar to Vion et al., Bazzi and Anderson investigated apoptosis in the absence of IFT88. In line with the results of Vion et al., Bazzi and Anderson could not measure altered apoptosis in *Ift88*^−/−^ mouse embryos [41]. Interestingly, Bazzi and Anderson elucidated that apoptosis is increased in mutants that lack both primary cilia and centrosomes suggesting a regulation of apoptosis by the centrosome rather than by primary cilia. In contrast to this hypothesis, three reports demonstrated that IFT88 deficiency leads to elevated apoptosis in developing zebrafish eyes [42,43,44]. To date, the reason for this discrepancy is unclear but is likely due to cell type-specific differences. Furthermore, the mentioned studies performed experiments in different species. While Vion et al. examined avians and Bazzi and Anderson performed investigations in mice, the increase in apoptosis caused by IFT88 deficiency was found only in zebrafish [41,42,43,44,45]. Consequently, the observed differences in apoptosis measured in different IFT88-deficient organisms could also be traced back to differences in the relationship between primary cilia and cell death across different species. In addition to *Ift88*, other genes encoding members of the IFT-B complex (*Ift52*, *Ift57*, and *Ift172*) and *Ift*-associated genes (*elipsa*) have been knocked down or even inactivated in zebrafish. In these mutants, apoptosis in the zebrafish eye was quantified. These quantifications yielded the detection of an increased apoptosis in the eyes of zebrafish *Ift* mutants [42,43,44]. Importantly, loss of IFT172 in mouse embryos results in increased apoptosis in the developing brain [46]. Considering that an IFT172 deficiency caused elevated levels of apoptosis in zebrafish and mice [42,46] and that a *Tg-Cre;Ift88*^flox/flox^ mouse model (with thyrocyte-specific loss of primary cilia) showed increased apoptosis of thyrocytes [47], the above-mentioned species-specific relationship between primary cilia and cell death seems more unlikely than a cell type-specific relationship between primary cilia and cell death. However, the limited data require additional experiments to be performed in order to get a clearer picture. Further insights into the link between IFT proteins and apoptosis were provided by studies that overexpressed the IFT-B complex subunits IFT46 and IFT88. The overexpression of IFT46 in zebrafish embryos induced excessive apoptosis in the central nervous system and the overexpression of IFT88 in HeLa cells caused apoptosis in 40% of the overexpressing cells [48,49]. These data demonstrate that any IFT abnormality provokes an increase in apoptotic cell death. So far, the underlying mechanism of this phenomenon is unknown.

## 3. Ciliary Base Proteins and Cell Death

To shed further light on the relationship between primary cilia and apoptosis, it might be useful to focus on other ciliary proteins than the IFT proteins. In this context, we suggest investigating proteins that localise to the ciliary base. Schock et al. revealed that apoptosis is unchanged in *talpid^2^* (*ta^2^*) mutant avian embryos suffering from an orofaciodigital syndrome-like ciliopathy [50]. However, Dvorak and Fallon detected reduced cell death in *ta^2^* mutant avian embryos [51]. Since the investigations of Schock et al. were limited to cranial neural crest cells and the analyses of Dvorak and Fallon refer to limb buds and limbs, cell type-specific differences should be taken into account in the deciphering of the relationship between primary cilia and cell death. Mutations in C2 Domain-Containing Protein 3 (C2CD3) have been shown to be the causal genetic lesion for the avian *ta^2^* mutant [52]. C2CD3 localises to the distal appendages that anchor the BB to the cell membrane [53,54]. It is required for ciliogenesis, for recruiting proteins to the BB, for removing protein from the BB, for the docking of vesicles to the BB, for distal appendage assembly, for controlling centriole elongation, and for regulating HH signalling in a tissue-specific manner [54,55,56,57]. In addition to the BB protein C2CD3, some proteins that localise to the TZ are associated with cell death. Several studies revealed that the deficiency of the MKS/B9 module components Tectonic-2 (TCTN2), Tectonic-3 (TCTN3), and Transmembrane Protein 67 (TMEM67 alias MKS3) in mice and rats increases apoptosis. In *Tctn2*-negative mouse embryos, increased apoptosis was detected in the ventral neuroectoderm and facial ectoderm [58]. Loss of TCTN3 engenders raised apoptosis in the brain of murine embryos [59]. In *Tmem67* mutant rats, elevated apoptosis was found in the eyes [60]. In the TZ, TCTN2 ensures the proper amount of TMEM67 [61]. Previously, it was shown that TCTN2, TCTN3, and TMEM67 control ciliary gating and, hence, ciliary protein composition [62]. During the investigations focussing on the link between TCTN2, TCTN3 and apoptosis, it was hypothesised that these proteins negatively regulate apoptosis by controlling HH signalling. In both *Tctn2*^−/−^ and *Tctn3*^−/−^ mouse embryos, HH signalling was reduced [58,59]. Importantly, the decrease in Patched-1 (PTCH1), a receptor of the HH ligand and a negative regulator of HH signalling, within *Tctn2*-negative mouse embryos rescued apoptosis in the ventral neuroectoderm and facial ectoderm whereby verifying the hypothesis [58]. Independently of the studies involving *Tctn* mutant mice, Aoto and Trainor revealed a regulation of apoptosis by PTCH1 via controlling CASP9 activity [63]. Mechanistically, the association of PTCH1 with X-linked Inhibitory Apoptosis Protein (XIAP) is of great importance and both proteins localise to primary cilia. To our knowledge, XIAP is the only protein involved in the known cell death signalling pathways that localises to primary cilia. In the absence of the HH ligand, PTCH1 separates from XIAP, and XIAP leaves the cilium. In this state, the C-terminal domain of PTCH1 is processed in a CASP9-dependent manner, which results in mitochondrial dysfunction and apoptosis. In the presence of the HH ligand, it binds to PTCH1 which subsequently exits the cilium and gets degraded [63]. Combining these results with the above-mentioned studies, TCTN2 and most likely TCTN3 as well might affect apoptosis by regulating HH signalling. However, to date it is unknown where TCTN2 and TCTN3 act in the HH signalling pathway and how these proteins affect apoptosis. Following the results of Aoto and Trainor, it could be assumed that the amount of PTCH1 might be higher in the *Tctn2*^−/−^ and the *Tctn3*^−/−^ state. However, Sang and colleagues did not detect an altered PTCH1 amount in *Tctn2*^−/−^ mouse embryonic fibroblasts (MEFs) (Sang et al., 2011), but they showed that TCTN2 is crucial for the processing and function of the HH signalling mediator Glioma-Associated Oncogene Family Zinc Finger 3 (GLI3) (Sang et al., 2011). Full length GLI3 either functions as a transcriptional activator or, after getting processed by partial proteolysis, acts as a repressor that regulates HH target gene expression [64]. Previous work elucidated an involvement of GLI3 in the regulation of apoptosis [65,66,67]. This raises the intriguing question if TCTN2 and TCTN3 might govern apoptosis via GLI3. In order to approach this question, we suggest analysing other TZ proteins involved in the regulation of GLI3 processing and function. As a prominent example, RPGRIP1L functions as a component of the NPHP1-4-8 module and controls the processing of GLI3 by regulating proteasomal activity at the ciliary base [68]. Moreover, it was suggested that RPGRIP1L governs the amount of the GLI3 activator [69]. Contrary to the situation in the absence of TCTN2 and TCTN3, loss of RPGRIP1L does not result in altered apoptosis in mouse embryonic hearts, in mouse embryonic hypothalami, or in the mouse adult central nervous system [69,70]. Subject to the possible existence of a cell type-specific relationship between primary cilia and apoptosis, these data indicate that TCTN2 and TCTN3 do not govern apoptosis by regulating GLI3. Furthermore, loss of TCTN2 and TCTN3 results in a reduction of Smoothened (SMO), another mediator of HH signalling which functions upstream of GLI3, reflecting impaired ciliary gating [62]. Interestingly, although RPGRIP1L also governs ciliary gating, SMO enters the cilium properly in the absence of RPGRIP1L [9,68]. This difference might be related to the different effects of TCTN2/TCTN3 and RPGRIP1L on apoptosis. In endothelial cells, it was elucidated that SMO negatively regulates apoptosis whereby supporting this hypothesis [71]. However, this line of argument seems to be countered by the fact that TMEM67 deficiency does not alter the ciliary SMO amount, although mutation of TMEM67 leads to enhanced apoptosis in the eyes of rats [60,62]. Considering that the connecting cilium in photoreceptors differs from the cilia of other cells, the SMO hypothesis has to be carefully tested by including cell type-specific effects into the experimental design [15]. Recently, it was shown that the TZ also functions as a gatekeeper for ciliary exit of retrograde IFT trains [72]. In this context, it is not surprising that TCTN2 ensures the proper ciliary localisation of IFT88 as TCTN2 is deeply involved in the regulation of ciliary gating [61,62]. Since both the loss of TCTN2 and the loss of IFT88 can result in increased apoptosis [42,43,44,58], it is conceivable that TCTN2 governs cell survival via IFT88. However, the proper localisation of IFT88 to cilia also depends on RPGRIP1L [61] but—as mentioned before—loss of RPGRIP1L does not provoke enhanced apoptosis [69,70]. Moreover, IFT88 (component of the IFT-B complex) and IFT139 (component of the IFT-A complex) localise independently of TCTN1, another TZ protein that controls ciliary gating [62]. Interestingly, depletion of TCTN1 promotes apoptosis [73,74]. According to these data, it seems unlikely that TZ proteins regulate apoptosis via ensuring the proper ciliary localisation of IFT proteins. Another possibility for how TCTN3 controls apoptosis is its physical interaction with Nephrocystin 1 (NPHP1), another component of the NPHP1-4-8 module. Depletion of NPHP1 led to elevated apoptosis and, in the absence of TCTN3, the amount of NPHP1 is decreased in mouse embryonic brains [59,75]. Interestingly, it has been suggested that NPHP1 regulates apoptosis in cooperation with Polycystin 1 (PC1), but the mechanism of this regulation remains elusive [75]. PC1 deficiency causes polycystic kidney disease, and elevated apoptosis is one feature of polycystic kidney disease [76]. Although apoptosis is unaltered in the absence of RPGRIP1L, it was reported that RPGRIP1L deficiency reduces the ciliary amount of NPHP1 querying a potential regulation of apoptosis by TCTN3 via NPHP1 [12,70]. To conclude, several ciliary base proteins have an impact on apoptosis, but the mechanism underlying the regulation of apoptosis by these proteins needs to be elucidated. We suggest regulation via HH signalling (especially via the HH signalling components PTCH1, GLI3, and SMO), via IFT proteins or via NPHP1. In every case, there are arguments for and against this hypothesis. To complete the puzzle of how TZ proteins act to control apoptosis, more research is required.

## 4. Does the Number of Primary Cilia Correlate with the Apoptosis Rate?

Apart from IFT and TZ proteins, several other ciliary proteins affect apoptosis. Cell Division Cycle 42 (CDC42) takes part in targeting vesicles to primary cilia that contain proteins destined to be present within or at the cilium. Loss of this protein causes polycystic kidney disease in mice, decreases ciliogenesis, and increases apoptosis in mouse embryonic kidneys [77]. Furthermore, Galectin 3 (GAL3) deficiency in mice results in fewer ciliated chondrocytes and elevated chondrocyte apoptosis [78]. Since the deficiency of several ciliary proteins such as CDC42, GAL3, IFT88, and TCTN2 provokes a decreased number of primary cilia and concomitantly an increased apoptosis rate [42,43,44,58,62,77], it is tempting to speculate that the reduction of cilia numbers leads to elevated apoptosis and, hence, that primary cilia function as the cell’s life saver. However, there are examples in which the situation is different. Ribosomal RNA Processing 7 Homolog A (RRP7A) localises to primary cilia, and mutations in the gene encoding RRP7A cause primary microcephaly—a disorder characterised by decreased brain size and intellectual disability. Depletion of RRP7A results in a higher ciliation frequency and elevated apoptosis in zebrafish [79]. Centromere Protein J (CENPJ) governs ciliary disassembly and loss of CENPJ results in a higher number of primary cilia and in the development of a microcephaly. Furthermore, apoptosis is enhanced in the developing mouse cerebral cortex in the absence of CENPJ [80]. These examples demonstrate that the number of primary cilia does not correlate with the apoptosis rate. This conclusion is also supported by another study in which it was shown that Anoctamin 6 (ANO6 alias TMEM16F) localises to primary cilia. Although ANO6 deficiency did not affect ciliary formation, apoptosis was reduced in *Ano6*-deficient Madin–Darby Canine Kidney (MDCK) cells [81].

## 5. Conclusions

In this review, we discuss the relationship between primary cilia and cell death in vertebrate embryonic development. Several IFT and ciliary base proteins control apoptosis, but the mechanism of this regulation is yet unknown (Table 1). Based on studies focussing on the regulation of apoptosis by TZ proteins, we hypothesise that ciliary proteins might govern apoptosis via the HH signalling pathway. Moreover, the number of cilia could affect apoptosis, but a correlation between the number of cilia number and the apoptosis rate cannot be established.

Without doubt, ciliary proteins have an effect on apoptosis as their depletion results in an altered apoptosis rate. However, at present, it is difficult to make a clear statement about whether ciliary proteins govern apoptosis in a positive or negative fashion and about how ciliary proteins regulate apoptosis mechanistically. Potentially, the answers to these open questions might be hampered by cell type-specific or even species-specific differences in the control of apoptosis by ciliary proteins. For these reasons, future studies on the relationship between primary cilia and cell death should be very broad in scope, using different species and analysing numerous cell types. To get further insights into the link between cilia and cell death, it is essential to identify the type of cell death that is regulated by ciliary proteins. While activated CASP3 represents a marker specific to apoptosis [82], the detection of fragmented DNA by using a terminal deoxynucleotidyl transferase dUTP nick end labelling (TUNEL) assay does not determine a difference between apoptosis and ACD [83]. Accordingly, different methods should be used in future studies in order to specify the kind of cell death controlled by proteins localising at primary cilia.

## Figures and Tables

**Figure 1 jdb-10-00052-f001:**
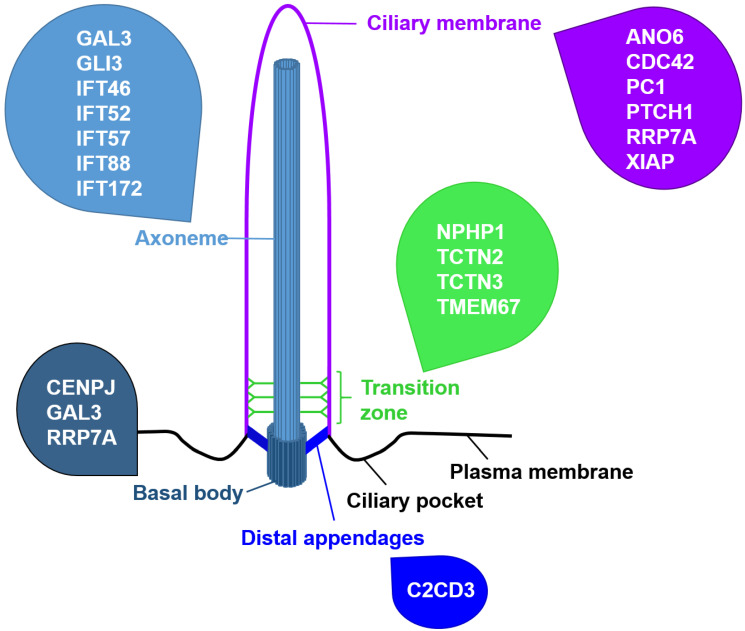
Schematic ciliary structure and subciliary localisation of proteins regulating cell death. Cell death-governing proteins are present at nearly every region of the cilium.

**Figure 2 jdb-10-00052-f002:**
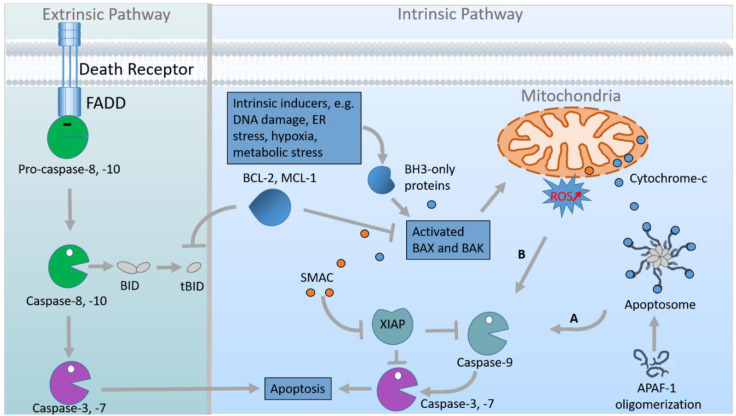
Extrinsic and intrinsic apoptosis-inducing signalling pathways. The extrinsic apoptotic pathway (**left panel**) is induced upon binding of a cognate ligand to the corresponding transmembrane death receptor such as the Tumour Necrosis Factor (TNF)-Related Apoptosis-Inducing Ligand (TRAIL) to its receptor (TRAILR). This results in activation of pro-caspases (Pro-Caspase-8 and Pro-Caspase-10, green pacman) through dimerisation mediated by adaptor proteins such as the FAS-Associated Death Domain Protein (FADD). Active Caspase-8 and Caspase-10 subsequently process and activate Caspase-3 and Caspase-7 (purple pacman) to induce apoptosis. The intrinsic pathway of apoptosis (**right panel**) is subdivided into apoptosome-dependent apoptosis (A) and apoptosome-independent apoptosis (B). Apoptosome-dependent apoptosis requires different intrinsic inducers (blue box upper left) that engage BCL-2 Homology Domain 3 (BH3)-only protein activation to activate BAX and BAK. Activated BAX and BAK induce permeabilisation of the mitochondrial outer membrane. Anti-apoptotic BCL-2 family proteins (e.g., BCL-2 and MCL-1) counteract this process. Subsequently, mitochondrial intermembrane proteins (Second Mitochondria-Derived Activator of Caspases (SMAC) and Cytochrome-c) are released into the cytosol, and then Cytochrome-c binds to Apoptotic Protease Activating Factor 1 (APAF-1) which assembles to the apoptosome and activates Caspase-9. Apoptosome-independent apoptosis is triggered by increasing ROS levels, which results in APAF-1-independent Caspase-9 activation. Active Caspase-9 activates Caspase-3 and Caspase-7 to induce apoptosis. Mitochondrial release of SMAC facilitates apoptosis by blocking the caspase inhibitor X-linked Inhibitor of Apoptosis Protein (XIAP). Caspase-8 cleavage of the BH3-only protein BH3-interacting domain death agonist (BID) enables crosstalk between the extrinsic and intrinsic apoptotic pathways. tBID, truncated BID; ER, endoplasmic reticulum; MCL1, myeloid cell leukaemia 1; tBID, truncated BID; ROS, reactive oxygen species.

**Table 1 jdb-10-00052-t001:** Overview of ciliary proteins involved in the regulation of cell death.

Name of the Protein	Impact on Apoptosis
ANO6	*Ano6*-deficient renal epithelial cells display reduced apoptosis [81]
C2CD3	Limb buds and limbs of *talpid^2^* (*ta^2^*) mutant avian embryos present reduced apoptosis [51]
CDC42	*Cdc42*^−/−^ mouse embryonic kidneys exhibit elevated apoptosis [77]
CENPJ	CENPJ-deficient developing mouse cerebrum cortices display increased apoptosis [80]
GAL3	Loss of GAL3 leads to enhanced chondrocyte apoptosis [78]
GLI3	GLI3 regulates apoptosis in the developing neural tube, face, and limb buds of mice [65,66,67]
IFT46	Overexpression of IFT46 in zebrafish embryos induces excessive apoptosis in the central nervous system [49]
IFT52	*Ift52* mutant zebrafish show an increased apoptosis of photoreceptors in the eyes [44]
IFT57	*Ift57* mutant zebrafish show an increased apoptosis of photoreceptors in the eyes [42]
IFT88	*Ift88*-deficient zebrafish retinas exhibit increased apoptosis [43]; *Ift88*^−/−^ zebrafish display photoreceptor apoptosis [42]; *Ift88*-deficient zebrafish ears show increased cell death [44]; *Tg-Cre;Ift88*^flox/flox^ mice suffering from thyrocyte-specific loss of primary cilia display increased apoptosis of thyrocytes [47]; overexpression of IFT88 in HeLa cells caused apoptosis in 40% of the overexpressing cells [48]
IFT172	*Ift172* mutant zebrafish show an increased apoptosis of photoreceptors in the eyes [42]
NPHP1	Depletion of NPHP1 in MDCK cells led to elevated apoptosis and cystic kidneys of patients with mutations in NPHP1 exhibit elevated apoptosis [75]
PC1	Overexpression of PC1 protects against apoptosis stimulation [76]
PTCH1	PTCH1 promotes apoptosis in a cell type-specific manner by regulating CASP9 activity [63]
RRP7A	Loss of RRP7A in zebrafish results in enhanced apoptosis [79]
SMO	SMO negatively regulates apoptosis in endothelial cells [71]
TCTN2	*Tctn2*-negative mouse embryos display increased apoptosis in the ventral neuroectoderm and facial ectoderm [58]
TCTN3	Loss of TCTN3 results in increased apoptosis in the brain of murine embryos [59]
TMEM67	*Tmem67* mutant rats show elevated apoptosis in the eyes [60]
XIAP	By binding to the C terminus of PTCH1, XIAP mediates its apoptosis-promoting function [63]

## Data Availability

Not applicable.

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
