# Peer review of "Life-Saver or Undertaker: The Relationship between Primary Cilia and Cell Death in Vertebrate Embryonic Development"

_jdb, 2022, doi:10.3390/jdb10040052_

Round 1

Reviewer 1 Report

In this manuscript, the authors review the current knowledge on the potential involvement of vertebrate cilia in the regulation of cell death. This is a very interesting and relevant topic, as cilia are being increasingly recognized as key players in many cellular and developmental processes. Moreover, as the authors mention, there is a growing number of diseases being associated with cilia malfunction. Therefore, understanding the biology of these organelles is fundamental to understand the cellular and molecular mechanisms underlying the development of these diseases.

I think the manuscript is very well written. Nevertheless, I would suggest adding a bit more information and discussion on specific points. Since a lot of the information concerns the transition zone, it would be relevant to provide more information about this ciliary domain, more specifically its important function as part of the ciliary gate that really defines and regulates the molecular composition of the ciliary compartment. Another point I believe to be very relevant is the structural and functional diversity of vertebrate cilia (motile and primary). The authors mention primary cilia as a general term, but in reality, vertebrates present very specialized cell and tissue-specific types of primary cilia. This is important, to discuss the findings presented in the paper, but also when we think that certain ciliopathy mutations, for example in TZ proteins, affect cilia in a cell/tissue-specific way.

In conclusion, I think this revision manuscript to be very relevant and I recommend it for publication, if the authors add a little more information and discussion about vertebrate cilia diversity, and the function of the TZ, and how these two aspects are relevant for this cilia link to cell death pathways. A nice figure putting together and summing up the discussed finds would also be nice.

Author Response

Dear Reviewer #1,

We deeply appreciate your comments on our manuscript entitled “Life-saver or undertaker: The relationship between primary cilia and cell death in vertebrate embryonic development” (Manuscript jdb-2031466). Your suggestions have significantly strengthened the manuscript. As detailed below we have addressed your specific comments in a point-by-point manner. Your questions/comments are written in italic letters, our answers not:

In this manuscript, the authors review the current knowledge on the potential involvement of vertebrate cilia in the regulation of cell death. This is a very interesting and relevant topic, as cilia are being increasingly recognized as key players in many cellular and developmental processes. Moreover, as the authors mention, there is a growing number of diseases being associated with cilia malfunction. Therefore, understanding the biology of these organelles is fundamental to understand the cellular and molecular mechanisms underlying the development of these diseases.

I think the manuscript is very well written. Nevertheless, I would suggest adding a bit more information and discussion on specific points. Since a lot of the information concerns the transition zone, it would be relevant to provide more information about this ciliary domain, more specifically its important function as part of the ciliary gate that really defines and regulates the molecular composition of the ciliary compartment.

Another point I believe to be very relevant is the structural and functional diversity of vertebrate cilia (motile and primary). The authors mention primary cilia as a general term, but in reality, vertebrates present very specialized cell and tissue-specific types of primary cilia. This is important, to discuss the findings presented in the paper, but also when we think that certain ciliopathy mutations, for example in TZ proteins, affect cilia in a cell/tissue-specific way.

We thank Reviewer #1 for pointing this out. As a consequence, we did not only provide more information about the transition zone and its function in ciliary gating (lines 35-42 and 272-276 and 283-294) as well as about specialised cell type-specific primary cilia types (lines 42-49), but we also inserted a new point about ciliary gating, SMO and its potential role in the regulation of apoptosis in consideration of the structural and functional diversity of primary cilia (lines 276-283).

In conclusion, I think this revision manuscript to be very relevant and I recommend it for publication, if the authors add a little more information and discussion about vertebrate cilia diversity, and the function of the TZ, and how these two aspects are relevant for this cilia link to cell death pathways. A nice figure putting together and summing up the discussed finds would also be nice.

We appreciate the idea of Reviewer #1 to insert a figure that summarises all discussed findings. However, we think that this figure would become confusing due to all the different data. For this reason, we created a table to give an overview about the cell death-regulating ciliary proteins and their impact on apoptosis (see Table 1).

Author Response

Dear Reviewer #2,

We deeply appreciate your comments on our manuscript entitled “Life-saver or undertaker: The relationship between primary cilia and cell death in vertebrate embryonic development” (Manuscript jdb-2031466). Your suggestions have significantly strengthened the manuscript. As detailed below we have addressed your specific comments in a point-by-point manner. Your questions/comments are written in italic letters, our answers not:

  1. Line 54-56: The authors should add reference to the statement. Also, authors should list a few primary cilia-mediated signaling pathways that participate in the mentioned cellular processes (during embryogenesis).

In the revised manuscript, we listed several cilia-mediated pathways regulating the enumerated cellular processes involved in embryonic development and added the associated references (lines 76-78).

  1. Line 69-76: The authors should be consistent in the format of protein (full names with the initial letter capitalized or not.) For example: TNF-related apoptosis-including ligand (TRAIL) is not capitalized, but Death Domain-Containing protein (FADD) with the first letters capitalized. This applies to the whole text.

We thank Reviewer #2 for mentioning the format error. The error is now corrected in the whole manuscript.

  1. Line 86, 87: cytochrome C should be cytochrome-c?

The name was revised accordingly.

  1. Line 126: the authors should introduce the specific function of C2CD3 protein and the signaling pathway that are involved.

We added information about the function of C2CD3 and about the signalling pathway which is regulated by C2CD3 (lines 213-226).

  1. Par 3. TZ proteins and cell death: The authors discussed the potential mechanisms of how TZ proteins deficiency regulates apoptosis. I suggest authors discuss three major hypothesizes in three paragraphs. First paragraph: In mouse embryos, HH decreases PTCH1; PTCH1 actives CASP9 to induce apoptosis. Second paragraph: TCTN2-GLI3; GLI3 regulates apoptosis; Third paragraph: TCTNsNPHP1-PC1 pathway.

Reviewer #2 is entirely correct. In order to provide a better overview, we summarised the different potential mechanisms underlying the regulation of apoptosis by the mentioned TZ proteins at the end of the section (lines 303-310).

  1. Line 236: make sure the following statement is correct: the number of primary cilia is not crucial for the apoptosis rate. Based on the authors description, cilia number is important for the apoptosis levels.

We thank Reviewer #2 for drawing our attention to this mistake. Of course apoptosis rate is often changed in case of an altered ciliary frequency (maybe with the exception of the mentioned ANO6 deficiency status). However, one cannot conclude from the data of various studies that the number of primary cilia correlates with the apoptosis rate. In some cases, less cilia appear together with an increased apoptosis, in other cases, more cilia and an increased apoptosis are observed. Consequently, we revised the sentence “These examples demonstrate that the number of primary cilia is not crucial for the apoptosis rate.“. Now it is as follows: “These examples demonstrate that the number of primary cilia does not correlate with the apoptosis rate.“ (lines 329 and 330).

  1. In the discussion, authors should make a short summary to each aspect, IFTs, TZ, and cilia numbers; what’s their potential roles in regulating apoptosis during embryogenesis?

In the new version of the manuscript, a short summary is present at the beginning of the „Conclusions“ section (lines 336-341).

Reviewer 3 Report

This manuscript reviews the relationship between primary cilia and cell death which is a topic largely unknown.

Specific comments

Page 2 lines 48-53. I do not agree with the statement “the development of more frequent diseases such as cancer or neurodegenerative diseases is related to primary cilia”. Which are the examples demonstrating this statement?

Concerning the intrinsic pathway of apoptosis, the authors did not mention in their text or in figure 2 the apoptosome independent cell death which has been described and should be added for completion (Indrieri et al., EMBO Mol Med 2013)

It is not clear to me why the transition zone was selected as ciliary section not related to IFT to discuss the relationship between primary cilia and apoptosis. The rationale should be explained. What about basal-body proteins?

Figure 2

What is the meaning of the underscore sign by the cytochrome c?

MCL1 is indicated in the text as MCL-1. The symbols should be the same

I would suggest for clarity to mention BCL-XL also in the text

Author Response

Dear Reviewer #3,

We deeply appreciate your comments on our manuscript entitled “Life-saver or undertaker: The relationship between primary cilia and cell death in vertebrate embryonic development” (Manuscript jdb-2031466). Your suggestions have significantly strengthened the manuscript. As detailed below we have addressed your specific comments in a point-by-point manner. Your questions/comments are written in italic letters, our answers not:

This manuscript reviews the relationship between primary cilia and cell death which is a topic largely unknown.

Specific comments

Page 2 lines 48-53. I do not agree with the statement “the development of more frequent diseases such as cancer or neurodegenerative diseases is related to primary cilia”. Which are the examples demonstrating this statement?

We added some examples to support our statement (lines 65-74).

Concerning the intrinsic pathway of apoptosis, the authors did not mention in their text or in figure 2 the apoptosome independent cell death which has been described and should be added for completion (Indrieri et al., EMBO Mol Med 2013)

We thank Reviewer #3 for mentioning this point and added the apoptosome-independent cell death to Figure 2 and to the text (lines 117-118 and 137-139).

It is not clear to me why the transition zone was selected as ciliary section not related to IFT to discuss the relationship between primary cilia and apoptosis. The rationale should be explained. What about basal-body proteins?

We appreciate this comment of Reviewer #3 and discussed links between TZ proteins and IFT proteins in the revised version of the manuscript (lines 283-294). In regard to basal body proteins, we found a relationship between C2CD3 and apoptosis. To provide more clarity, we shifted the part about C2CD3 from the IFT section to the section about ciliary base proteins (lines 213-226). Other proteins such as GAL3 or RRP7A do not only localise to the BB but also to other subciliary locations.

Figure 2

What is the meaning of the underscore sign by the cytochrome c?

We removed the underscore in Figure 2.

MCL1 is indicated in the text as MCL-1. The symbols should be the same

We revised the symbol.

I would suggest for clarity to mention BCL-Xalso in the text

We removed BCL-XL due to species-specific differences.